# Contribution Ratio of Metatarsal Osteotomy and First Tarsometatarsal Joint Reduction in Moderate to Severe Hallux Valgus Correction

**DOI:** 10.3390/ijerph19148243

**Published:** 2022-07-06

**Authors:** Dong-Kyo Seo, Hasung Park, Myeong Geun Song, Youngjoo Jung, Young-Rak Choi

**Affiliations:** 1Department of Orthopedic Surgery, Gangneung Asan Hospital, College of Medicine, University of Ulsan, Gangneung-si 25440, Korea; dongkyoseo@gmail.com (D.-K.S.); hasungsmile@gmail.com (H.P.); jepi0999@gmail.com (M.G.S.); cdcd1425@gmail.com (Y.J.); 2Department of Orthopedic Surgery, Asan Medical Center, University of Ulsan, College of Medicine, Seoul 05505, Korea

**Keywords:** hallux valgus, intermetatarsal angle, translation, tarsometatarsal joint

## Abstract

Hallux valgus is a common foot and ankle disease, for which numerous surgical procedures were introduced. So, understanding the mechanism of deformity reduction is important to select the proper method. Intermetatarsal angle (IMA) determines the severity of hallux valgus, which is influenced by the translated metatarsal head and the reduction of the first tarsometatarsal joint. We hypothesized that both of the mechanisms simultaneously contribute to the correction of IMA. Hallux valgus (70 feet) operated with a Scarf osteotomy with the Akin procedure were reviewed. Hallux valgus angle (HVA), IMA (mechanical and anatomical), hallux valgus interphalangeal angle (HVIP), distal metatarsal articular angle (DMAA), and sesamoid position were checked. The ratio of contributions to the IMA changes were calculated and compared. When the individual contributions by metatarsal head translation and first tarsometatarsal joint reduction were compared, metatarsal head translation contributed by 82%, whereas first tarsometatarsal joint reduction contributed by 18%. Both were responsible for mechanical IMA correction. However, IMA change by metatarsal head translation was a major correction mechanism compared to anatomical IMA change by first tarsometatarsal joint reduction.

## 1. Introduction

Hallux valgus is a common disease of the foot and ankle with a prevalence of about 23.0% to 35.7% [1,2,3]. Numerous surgical procedures were introduced in the literature [4,5,6,7]. So, selecting the proper surgical method is important to achieve an acceptable outcome. However, there is no consensus as to which is the gold standard for treating hallux valgus [4]. Therefore, diagnosis of the disease severity and understanding the mechanism of the deformity reduction is important for selecting the proper method [8].

The hallux valgus angle (HVA) and intermetatarsal angle (IMA) are the main targets for deformity correction. The HVA indicates an angle made by the central axis of the first metatarsal and proximal phalanx. The IMA indicates an angle between the central axis of the first and second metatarsal. The severity of the hallux valgus is usually determined by both HVA and IMA. The HVA is related to the valgus deformity of the first phalanx. The IMA is related to the width of the feet and the prominence of the bunion.

First metatarsal osteotomy with lateral translation of the metatarsal head is a common method for correcting the IMA. Various metatarsal osteotomies, having their own correcting power, were recommended, according to the severity of the hallux valgus [4]. The post-operative first metatarsal axis is made by a line connecting the base of the first metatarsal bone and the center of the first metatarsal head. The center of the metatarsal head is moved laterally by translation with osteotomy, which decreases the IMA. However, the corrected IMA is influenced not only by the translated metatarsal head but also by the reduction of the first tarsometatarsal (TMT) joint. The main correcting factor has not been identified, and the ratio of the individual contributions from the two mechanisms in the IMA correction has not been fully understood. We hypothesized that both a reduction in the first TMT joint and a translated first metatarsal head simultaneously contributes to the corrected IMA. So, we investigated the changes and ratio of two factors in hallux valgus correction.

## 2. Materials and Methods

The local institutional review board approved this study (GNAH 2021-11-011-001). Seventy feet that underwent hallux valgus operation with a Scarf osteotomy with the Akin procedure from March 2016 to January 2021 were reviewed.

The inclusion criteria were operated hallux valgus feet with moderate to severe deformity (pre-operative IMA ≥ 13, HVA ≥ 20). Hallux valgus feet with first metatarsal joint degenerative change, a previous fracture history of the first or second metatarsal, and patients with neuromuscular disease or diabetes mellitus were excluded.

The operation was conducted under spinal anesthesia with an inflated pneumatic tourniquet. The adductor hallucis tendon insertion and lateral capsule were completely released through a dorsal incision between the first and second metatarsal heads. After medial incision and bunionectomy, the first metatarsal head was translated laterally (about 5–7 mm translation) with a Scarf osteotomy. One or two 2.4 mm cortical screws were fixed from the dorsal to the plantar side. The Akin osteotomy was performed in all of the feet. The medial joint capsule was repaired in a slightly supinated position of the proximal phalanx of the first toe. The skin was sutured with 3.0 non-absorbable nylon.

From the second post-operative day, tolerable partial weightbearing walk was allowed with rigid sole shoes. A passive range of motion exercise of the first metatarsal joint was recommended. The rigid sole shoe was removed at post-operative 8 weeks. Weightbearing foot plain radiographs were checked at post-operative 3, 6, and 12 months.

Angular measurement was completed in the pre- and post-operative weightbearing foot anterior–posterior plain radiographs. The pre-operative images were taken within one month from operation day, and the post-operative images were taken at post-operative six months. The hallux valgus angle (HVA), intermetatarsal angle (IMA), hallux valgus interphalangeus angle (HVIP), distal metatarsal articular angle (DMAA), and sesamoid position were checked (Figure 1) [9].

The IMA was classified into two subtypes, mechanical and anatomical. The IMA was defined as the angle between the longitudinal axis of the first and second metatarsals. In pre-operative plain radiographs, both the mechanical and anatomical IMAs were the same. In the post-operative plain radiographs, a mechanical IMA was defined as the intersection angle between the central axis of the first distal metatarsal (a line connecting the base and articular center of the first metatarsal) and the second metatarsal (Figure 2).

An anatomical IMA was defined as the intersection angle between the longitudinal axis of the proximal first metatarsal and second metatarsal (Figure 3).

Mechanical IMA change was defined as the difference between the pre- and post-operative IMA. The post-operative mechanical IMA was defined as the sum of both angular changes made by the reduction of first TMT joint (anatomical IMA) and translated metatarsal head fragment. The proportions of anatomical IMA and metatarsal head translation, and related changes were calculated and compared. The American Orthopedic Foot and Ankle Society (AOFAS) Hallux Metatarsophalangeal–Interphalangeal score and Foot and Ankle outcome score (FAOS) were recorded at the last follow-up. A paired *t*-test was used to compare pre- and post-operative angular changes.

## 3. Results

Mean age was 57.2 ± 12.2 (range, 21–80) years, and the mean follow-up period was 9.1 ± 3.3 (range, 6.0–32.7) months. Other demographic details are described in Table 1.

The post-operative HVA, IMAs (both mechanical and anatomical), DMAA, and sesamoid position were decreased significantly (*p*-value < 0.001). The HVIP was increased (*p* < 0.001) (Table 2).

Both of the functional scores were improved significantly (*p* < 0.001). The mechanical IMAs were reduced in all of the feet. In addition, the mean anatomical IMA was decreased; however, 12 feet showed an increased anatomical IMA and 10 feet showed no change.

Among the mean IMA decrease of 11.2 degrees, the IMA change by metatarsal head translation and first TMT joint reduction was 8.5 and 2.7 degrees, respectively. In a comparison of the two factors, metatarsal head translation was responsible for 82.2% of the totally corrected IMA, whereas first TMT joint reduction contributed to only 17.8% (Table 3).

The mean AOFAS Hallux Metatarsophalangeal–Interphalangeal score and FAOS were significantly improved post-operatively (*p* < 0.001). There were no complications that needed revision surgery.

## 4. Discussion

IMA in hallux valgus is an important factor that defines the severity of this disease. It also correlates with the width of feet, the protrusion of bunion, and the severity of HVA. So, correcting the IMA is one of the most important procedures in treating hallux valgus. Numerous metatarsal osteotomies were proposed, according to the severity of the disease [10,11,12,13,14,15]. Scarf osteotomy is one of the most powerful and commonly used procedures, which has multiple variations for osteotomy sites [16,17,18]. We performed Scarf osteotomy in all of the patients combined with the Akin procedure.

An IMA is defined as the angle between the longitudinal axis of the first and second metatarsal bones. The longitudinal axis line connects the base of the metatarsal to the center of the head. After the operation, the center of the head positioned on the first metatarsal axis shifts laterally with translation of the distal bone fragment. However, the correction of the IMA is not only decided by the movement of the metatarsal head. The reduction of the first TMT joint results in lateral angulation of the proximal fragment with the first metatarsal. We believe that this mechanism is influenced by the correction of HVA and lateral soft-tissue release. Thus, post-operative IMA reduction (mechanical IMA) is achieved by both a lateral translation of the first metatarsal head and a reduction in the first TMT joint (anatomical IMA).

In our results of angular changes, all of the statuses, except HVIP, decreased significantly, which was corrected with an operation. The functional score also showed improved results. Preoperative HVIP would be underestimated with pronation of the first proximal phalanx in weight-bearing plain radiographs [19,20]. The distal articular surface of the first proximal phalanx is laterally deviated more than the proximal articular surface in the anterior–posterior view [21]. However, it would be checked smaller in a pronated position when the dorsal surface is shown as the medial cortex in plain radiographs. So, we hypothesized that the pre-operative HVIP would be checked smaller than the post-operative value. These phenomena need to be studied.

The mean mechanical IMA angle change was 11.2 (range, 2–18) degrees. The mean IMA change by metatarsal head translation was 8.5 degrees, whereas the mean anatomical IMA change (first TMT joint reduction) was 2.7 degrees. The proportion of metatarsal head translation and first TMT joint reduction was about 82% and 18%, respectively. Both factors were responsible for the post-operative mechanical IMA correction. However, the IMA change by metatarsal head translation was a major correction mechanism, compared to change by first TMT joint reduction.

We think that the proportion of reduced IMAs is decided by the amount of distal metatarsal fragment translation by osteotomy. With a powerful osteotomy such as Scarf, the first metatarsal head moves laterally as much as possible to the second metatarsal head after the distal fragment translation. Then, there would be less room for the first TMT joint reduction. However, with less powerful osteotomies such as the distal chevron, the first TMT reduction would have more proportion. Furthermore, with a modified McBride procedure with no metatarsal osteotomy, the IMA reduction would be made by the first TMT reduction only.

In other reports [22], the hallux valgus was managed with distal chevron osteotomy; both the mechanical and anatomical IMA decreased in the severe group (IMA > 16 degrees). However, the anatomical IMA was increased in the mild group (IMA < 11 degrees). They suggested that obstructing the lateral translation of the first metatarsal head can result in the medialization of the proximal fragment. In our results, the mean IMA was decreased post-operatively. However, minimal medialization (increased anatomical IMA) was seen in 12 feet (from zero to −5 degrees), and no change in 10 feet (zero degrees). The mean pre-operative IMA of increased or unchanged anatomical IMA feet was 15.2 ± 3.1 (range, 10–22), and the decreased anatomical IMA feet group was 18.4 ± 2.8 (range, 14–27). Relatively mild pre-operative IMA feet showed less anatomical IMA changes post-operatively. The difference between the two groups was significant (*p* < 0.001); however, both of the groups only had a moderate hallux valgus deformity (13 < pre-operative IMA < 20 degrees). Increased post-operative anatomical IMA was also observed in Chevron osteotomy in other reports [23]. We also believe that over-translation of the distal fragment could result in the medialization of the proximal fragment.

The importance of metatarsal osteotomy and translation of distal fragment for IMA correction was identified with our results. Therefore, selecting the proper osteotomy according to the severity of the deformity is important for correcting and preventing a recurrence. Our study has some limitations. This is a case series, and the ratio of metatarsal osteotomy and first TMT joint reduction was not compared with other osteotomies nor mild degree deformity. It should be compared in future studies.

## 5. Conclusions

Both metatarsal head translation and proximal fragment reduction were responsible for IMA correction. However, the IMA change by metatarsal head translation was a major correction mechanism, compared to an anatomical IMA change by first TMT joint reduction. Selecting a proper osteotomy is important to achieve an acceptable deformity correction.

## Figures and Tables

**Figure 1 ijerph-19-08243-f001:**
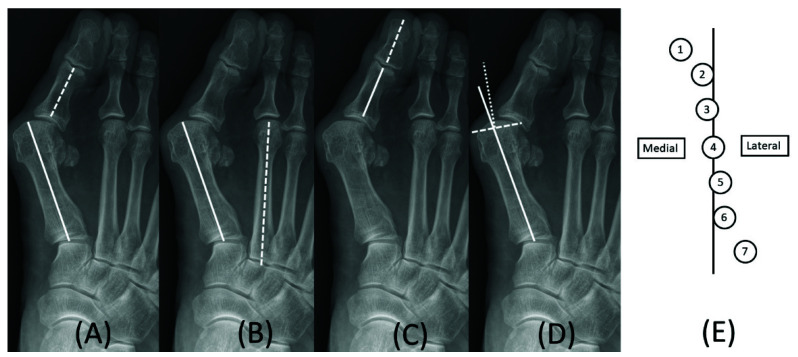
Preoperative angle measurements in a plain radiograph. Hallux valgus angle was defined as the intersection angle made by the longitudinal axis of the first metatarsal (solid) and proximal phalanx (dash) (**A**); The intermetatarsal angle was the intersection angle made by the longitudinal axis of the first (solid) and second (dash) metatarsal (**B**); Hallux valgus interphalangeal angle was the intersection angle made by the longitudinal axis of proximal (solid) and distal (dash) phalanx (**C**); Distal metatarsal articular angle was the intersection angle made by the longitudinal axis of the first metatarsal (solid) and bisecting line (dot) of the articular surface (dash) (**D**); The sesamoid position was evaluated with medial sesamoid position to longitudinal line of the first metatarsal (**E**).

**Figure 2 ijerph-19-08243-f002:**
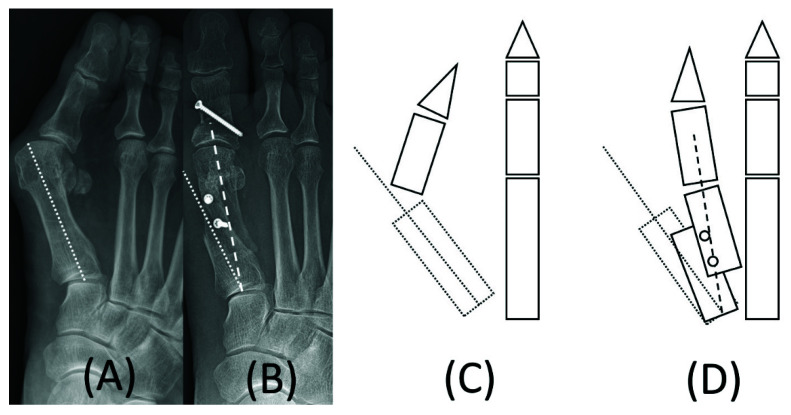
Mechanical axis of the first metatarsal in pre- and postoperative plain radiographs. Both mechanical and anatomical axes of the first metatarsal (dot line) are the same in pre-operative plain radiograph (**A**); The postoperative mechanical axis of the first metatarsal (dash line) is a connecting line to the base and articular center of the first metatarsal, which are laterally angulated compared to the pre-operative axis (**B**); In the schematic drawing, the postoperative mechanical axis (dash line) was corrected laterally (**D**) compared to the pre-operative first metatarsal (dot line) axis (**C**).

**Figure 3 ijerph-19-08243-f003:**
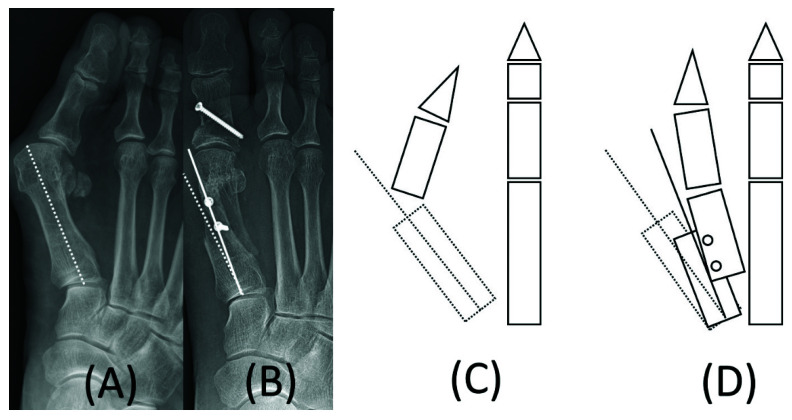
Anatomical axis of the first metatarsal in pre- and post-operative plain radiographs. Both mechanical and anatomical axis of the first metatarsal (dot line) are the same in pre-operative plain radiograph (**A**); The post-operative anatomical axis (solid line) is a longitudinal axis of the proximal first metatarsal (**B**); In the schematic drawing, the post-operative anatomical axis (solid line) was corrected laterally (**D**) compared to the pre-operative first metatarsal (dot line) axis (**C**).

**Table 1 ijerph-19-08243-t001:** Demographic status.

Feet	70
Age	57.2 ± 12.2
Follow up (months)	9.1 ± 3.3 (range, 6.0–32.7)
Male: Female	9:61
Side (Right to Left)	35:35
BMI	24.3 ± 2.9 (range, 18.8–30.9)

**Table 2 ijerph-19-08243-t002:** Comparison of radiologic angular changes.

Group	Preoperative	Postoperative	*p*-Value
Hallux valgus angle	33.3 ± 8.4 (20–49)	5.5 ± 6.4 (−10–22)	<0.001
Intermetatarsal angle (Mechanical)	17.5 ± 3.1 (13–27)	6.4 ± 2.9 (1–17)	<0.001
Intermetatarsal angle (Anatomical)	17.5 ± 3.1 (13–27)	14.8 ± 3.2 (4–27)	<0.001
Hallux valgus interphalangeal angle	4.9 ± 6.2 (−10–20)	10.2 ± 5.8 (0–22)	<0.001
Distal metatarsal articular angle	17.7 ± 9.3 (0–41)	6.1 ± 5.9 (−4–28)	<0.001
Sesamoid position	5.8 ± 0.9 (4–7)	2.6 ± 0.9 (1–5)	<0.001
^1^ AOFAS Hallux Metatarsophalangeal–Interphalangeal score	41.2 ± 11.2 (8–62)	90.8 ± 5.6 (82–100)	<0.001
^2^ FAOS	48.6 ± 7.4 (38–68)	83.4 ± 4.9 (71–92)	<0.001

^1^ AOFAS, The American Orthopedic Foot and Ankle Society; ^2^ FAOS, Foot and Ankle Outcome Score.

**Table 3 ijerph-19-08243-t003:** Contribution ratio of angular changes in mechanical and anatomical intermetatarsal angle.

Mechanical IMA Change	Degrees	Ratio (%)
by metatarsal head translation	8.5 ± 2.2 (3–15)	82.2
by first TMT joint reduction (Anatomical IMA change)	2.7 ± 2.8 (−5–9)	17.8
Total	11.2 ± 3.1 (2–18)	100

## Data Availability

Not applicable.

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
