# Peer review of "Contribution Ratio of Metatarsal Osteotomy and First Tarsometatarsal Joint Reduction in Moderate to Severe Hallux Valgus Correction"

_ijerph, 2022, doi:10.3390/ijerph19148243_

Round 1

Reviewer 1 Report

The article reports a study on 70 feet affected by allux valgus. The manuscript deals with an interesting topic with possibile clinical and surgical impication for a widespread feet deformity. It’s well written and structured, but some revisions are needed to make it worthy of publication:

-Introduction is clear and well structured, however, references are few and some additional insights could improve it;

-Materials and methods, as well as results, are complete. Tables are essential and properly structured;

-Discussion should be reorganized to make understanding more fluent and some references should be added;

-Conclusion drawn by tre authors are in agreement with tre results.

Author Response

The article reports a study on 70 feet affected by hallux valgus. The manuscript deals with an interesting topic with possibile clinical and surgical impication for a widespread feet deformity. It’s well written and structured, but some revisions are needed to make it worthy of publication:

-Introduction is clear and well structured, however, references are few and some additional insights could improve it;

#Answer

Thank you for your advice.

Following references were added.

Kaufmann, G., et al. (2019). "Minimally invasive versus open chevron osteotomy for hallux valgus correction: a randomized controlled trial." Int Orthop 43(2): 343-350.

Lenz, C. G., et al. (2021). "Scarf osteotomy for hallux valgus deformity: Radiological outcome, metatarsal length and early complications in 118 feet." Foot Ankle Surg 27(1): 20-24.

Perera, A. M., et al. (2011). "The pathogenesis of hallux valgus." J Bone Joint Surg Am 93(17): 1650-1661.

Ray, J. J., et al. (2019). "Hallux Valgus." Foot Ankle Orthop 4(2): 2473011419838500.

Smyth, N. A. and A. A. Aiyer (2018). "Introduction: Why Are There so Many Different Surgeries for Hallux Valgus?" Foot Ankle Clin 23(2): 171-182.

-Materials and methods, as well as results, are complete. Tables are essential and properly structured;

-Discussion should be reorganized to make understanding more fluent and some references should be added;

#Answer

Discussion section was edited more fluently and unnecessary paragraphs were deleted.

Following reference was added.

Chan, J. Y., et al. (2020). "Distal Chevron Osteotomy Increases Anatomic Intermetatarsal Angle in Hallux Valgus." Foot Ankle Orthop 5(4): 2473011420960710.

-Conclusion drawn by tre authors are in agreement with tre results.

Reviewer 2 Report

Hallux valgus as a common foot pathology, still attracts attention of the researches. I cannot agree with the introduction that IMA is the most important pathologic finding in the foot x-ray. Of course, severity of the deformity depends on it, but it is not the only "correcting target in hallux valgus operation". 

The inclusion criteria should be more underlined - authors chose only patients with IMA>13 deg. 

The way how "total IMA" change is measured is completely not understandable. I think, that mechanical IMA covers both changes (due to head translation and due to new position of proximal part of the first metatarsal. It is not clearly described. 

Author Response

Hallux valgus as a common foot pathology, still attracts attention of the researches. I cannot agree with the introduction that IMA is the most important pathologic finding in the foot x-ray. Of course, severity of the deformity depends on it, but it is not the only "correcting target in hallux valgus operation". 

# Answer

; Thank you for the advice. I agree with your opinion. It was my mistake that commenting IMA was the most important factor for correction. I corrected the paragraphs like following.

(Hallux valgus angle (HVA) and Intermetatarsal angle (IMA) is main target for deformity correction. HVA indicates an angle made by the central axis of first metatarsal and proximal phalanx. IMA indicates an angle between the central axis of the first and second metatarsal. The severity of hallux valgus is usually determined by both HVA and IMA. HVA is related with valgus deformity of first phalanx. IMA is related with width of feet and prominence of bunion.)

The inclusion criteria should be more underlined - authors chose only patients with IMA>13 deg. 

# Answer

; it was corrected (Inclusion criteria were operated hallux valgus feet with moderate to severe deformity (preoperative IMA ≥ 13, HVA ≥ 20).

The way how "total IMA" change is measured is completely not understandable. I think, that mechanical IMA covers both changes (due to head translation and due to new position of proximal part of the first metatarsal. It is not clearly described. 

# Answer

; Thank you for the comment. it was corrected.

(IMA change was defined as the difference between pre- and postoperative IMA. Postoperative mechanical IMA was defined as the sum of both angular changes made by reduction of proximal metatarsal fragment (anatomical IMA) and translated distal metatarsal head fragment. The proportions of anatomical IMA and metatarsal head translation, and related changes were calculated and compared.)

Reviewer 3 Report

The manuscript compare different therapeutic methods to conclude the best mechanism of reduction for deformity. It is well written. My only comment is the table 1 must be in results rather than the methods 

Author Response

The manuscript compare different therapeutic methods to conclude the best mechanism of reduction for deformity. It is well written. My only comment is the table 1 must be in results rather than the methods 

# Answer

; Thank you for the comment. Table 1 was moved to the result section.

Round 2

Reviewer 2 Report

I think, now  it can be published,